# Molecular Cloning of the *B4GALNT2* Gene and Its Single Nucleotide Polymorphisms Association with Litter Size in Small Tail Han Sheep

**DOI:** 10.3390/ani8100160

**Published:** 2018-09-20

**Authors:** Xiaofei Guo, Xiangyu Wang, Benmeng Liang, Ran Di, Qiuyue Liu, Wenping Hu, Xiaoyun He, Jinlong Zhang, Xiaosheng Zhang, Mingxing Chu

**Affiliations:** 1Key Laboratory of Animal Genetics and Breeding and Reproduction of Ministry of Agriculture, Institute of Animal Science, Chinese Academy of Agricultural Sciences, Beijing 100193, China; guoxfnongda@163.com (X.G.); xiangyu_wiggle@163.com (X.W.); 18812109228@163.com (B.L.); dirangirl@163.com (R.D.); qiuyue1983921@163.com (Q.L.); pinkyhoho@163.com (W.H.); hedayun@sina.cn (X.H.); 2Tianjin Institute of Animal Sciences, Tianjin 300381, China; jlzhang1010@163.com (J.Z.); zhangxs0221@126.com (X.Z.)

**Keywords:** *FecL* mutation, *B4GALNT2* gene, RACE, expression profile, litter size

## Abstract

**Simple Summary:**

In French Lacaune sheep, the *B4GALNT2* (beta-1, 4-*N*-acetyl-galactosaminyl transferase 2) gene was considered as the potential gene for a *FecL* (mutation), which regulates the ovine ovulation rate. Three specific mutation sites linked with the *FecL* mutation have not been previously found in 11 sheep breeds. However, two mutations of g.36946470C > T and g.36933082C > T in the exon of *B4GALNT2* were found to have had a significant effect on the litter size in the first parity for Small Tail Han (STH) Sheep (*p* < 0.05). *B4GALNT2*, which is mainly expressed in ovine ovary, also plays an important role in sheep reproduction. Furthermore, we discovered two transcription start sites (TSS) of *B4GALNT2* in its 5′-flanking region in ovine granule cells in vitro.

**Abstract:**

A new fecundity gene named the *FecL* (mutation), which regulates the ovulation rate, was discovered in French Lacaune sheep. The *B4GALNT2* (beta-1, 4-*N*-acetyl-galactosaminyl transferase 2) gene was considered as the potential *FecL* mutation gene. This study explores whether the effect of the *FecL* mutation exists in other sheep breeds, and the features of the *B4GALNT2* gene in terms of the molecular structure and its expression profile. Using Sanger sequencing, we found that high and low fecundity breeds from among 11 measured sheep breeds all had no variation in the three specific mutation sites, which were linked with the *FecL* mutation. However, two mutations of g.36946470C > T and g.36933082C > T in the exon of *B4GALNT2* had a significant effect on litter size in the first parity for Small Tail Han (STH) Sheep (*p* < 0.05). Two transcription start sites (TSS) of *B4GALNT2* in its 5′-flanking region were discovered in ovine granule cells in vitro, through the RACE (Rapid amplification of cDNA ends) method. Except for in the kidney and oviduct, no significant difference in expression levels had been found between STH sheep and Tan sheep breeds. The *B4GALNT2* gene, as a candidate for *FecL*, may have a relationship with the differences in litter size in STH sheep. *B4GALNT2* is mainly expressed in the ovine ovary, which also suggests that *B4GALNT2* plays an important role in sheep reproduction.

## 1. Introduction

On account of annual mutton production being determined by annual sheep slaughter rates and individual average meat yield, high levels of ewe reproduction are equally important to superior carcass traits in meat sheep production. However, reproduction traits with relatively low heritability that are generally not expressed until puberty, and are normally recorded only in females, resulting in the genetic improvement of reproduction in sheep being a challenge [1,2]. Litter size and the ovulation rate (OR), involved in reproduction traits, could be regulated by the action of single genes with major effects, called fecundity (Fec) genes [3,4]. For the last three decades, geneticists have created informative families for segregation studies and fine mapped some of the Fec genes that affect ovine litter size and OR [5]. To date, the most efficient of the three Fec genes, Bone Morphogenetic Protein-15 (*BMP15*) [6,7,8,9,10], Growth and Differentiation Factor-9 (*GDF9*) [10,11,12,13], and Bone Morphogenetic Protein receptor type-1B (*BMPR1B*) (the *FecB* gene [7,8,14,15,16]) have all belonged to the Bone Morphogenetic Protein (BMP) system [5]. Nevertheless, a newly founded *FecL* gene named beta-1, 4-*N*-acetyl-galactosaminyl transferase 2 (*B4GALNT2*), encoding for a glycosylation enzyme, which was not related to the BMP family, recently attracted the attention of researchers [17].

In the meat strain of the Lacaune sheep breed in France, large variations in litter size and OR have been observed, and at least two major genes could explain this variation. One was X-linked—namely *Fec*, and the second was autosomal—namely *FecL* [6,18,19,20]. Similarly to the *FecB* mutation, the influence of the *FecL* mutation on OR was also additive, with extra ovulations increasing by approximately 1.5 for one copy and by 3.0 for two copies [4,17,18,19]. In 2009, Drouilhet et al. reported that a unique haplotype was associated with the *FecL* mutation, and they had reached the conclusion that the DLX3:c. * 803A > G Single Nucleotide Polymorphisms (SNP) provided accurate classification of 99.5% of sheep as carriers or non-carriers of the *FecL* mutation [4]. In 2013, consideration of two mutations (the SNP g.36938224T > A mutation, localized in the intron 7 of *B4GALNT2*, and the SNP g.37034573A > G mutation, localized in the intergenic sequence between *B4GALNT2* and *EZR*) that were closely associated with the *FecL* mutation appeared to suggest that *B4GALNT2* appeared as the best positional and expressional candidate for the *FecL* gene [17]. The *B4GALNT2* gene in humans, encoding for the glycosylation enzyme beta-1, 4-*N*-acetyl-galactosaminyl transferase 2, was involved in the pathway of protein glycosylation or modification [21].

The *FecB* gene was found in Booroola Merino sheep and regarded as the first major gene for prolificacy, and was also identified in various other sheep breeds [7,22,23,24,25]. However, whether the effects of the *FecL* mutation, which was found in the Lacaune sheep breed, also exists in other sheep breeds remains unknown. In the present study, three specific linked marker mutations (*DLX3*:c. * 803A > G, g.36938224T > A, g.37034573A > G) for *FecL* were detected using the sequencing method in 11 sheep breeds (Small Tail Han (STH) sheep, Hu sheep, Cele Black sheep, Tan sheep, White Suffolk sheep, Black Suffolk sheep, East Friesian sheep, Dorset sheep, Mutton Merino sheep, Dorper sheep, and Corriedale sheep). Based on previous data from the whole-genome sequencing (WGS) performed by our research team, seven SNPs of the *B4GALNT2* gene were detected in 99 experimental sheep [26,27]; we explored the frequency of the seven SNPs in STH sheep and studied their relationship with litter size. Then, the *B4GALNT2* transcripts were cloned, and the corresponding protein sequence and its bioinformatics were predicted in the research. The tissue expression profiles of *B4GALNT2* in STH sheep and Tan sheep were implement, to explore the gene expression difference between high and low fecundity breeds. The data obtained, and the results of these analyses, will help us to understand the effect of the sheep *B4GALNT2* gene on sheep reproductive traits.

## 2. Materials and Methods

### 2.1. Experimental Animals and Sample Collection

High and low fecundity breeds were selected to detect the mutations with linkage with *FecL*. Jugular blood samples from 88 STH sheep (recorded with data of the litter size) and varied fecundity of different sheep breeds (STH sheep, Hu sheep, Cele Black sheep, Tan sheep, White Suffolk sheep, Black Suffolk sheep, East Friesian sheep, Dorset sheep, Mutton Merino, Dorper sheep, and Corriedale sheep). Samples from a total of 30 ewes, from these breeds, were collected and stored at −20 °C for DNA extraction. Three female STH sheep and three female Tan sheep, each aged two years old, were purchased from purebred herds from the same farm in the Ningxia province. The six selected sheep were healthy, similar in weight, and fed in an indoor setting under similar conditions of room temperature, illumination, feeding system, and nutrition level. The six female sheep were slaughtered in autumn when they accepted the teasing behavior for the advent of oestrum. The tissues of the heart, liver, spleen, lung, kidney, oviduct, uterine horn, uterine body, muscle, cerebellum, hypothalamus, pituitary, and ovary in estrus from the six sheep were collected and frozen in liquid nitrogen instantly, then stored at −80 °C for RNA extraction. All the experimental procedures mentioned in the present study were approved by the Science Research Department (in charge of animal welfare issues) of the Institute of Animal Sciences, Chinese Academy of Agricultural Sciences (IAS-CAAS) (Beijing, China). Ethical approval on animal survival was given by the animal ethics committee of IAS-CAAS (No. IASCAAS-AE-03, 12 December 2016).

### 2.2. Genotyping the Known SNPs Linked with the FecL Mutation

Genomic DNA were extracted from the blood samples with a Blood DNA kit (TIANGEN Biotech Co., Ltd., Beijing, China). The primers for the detection of two mutations (the mutation g.36938224T > A, localized in the intron 7 of *B4GALNT2*, and the mutation g.37034573A > G, localized in the intergenic sequence between *B4GALNT2* and *EZR*) and one linked marker mutation (*DLX3*:c. * 803A > G SNP) were designed based on the reference genome of ovis aries Oar_3.1 (Appendix A). In this study, a total of 11 sheep breeds (STH sheep, Hu sheep, Cele Black sheep, Tan sheep, White Suffolk sheep, Black Suffolk sheep, East Friesian sheep, Dorset sheep, Mutton Merino sheep, Dorper sheep, and Corriedale sheep) of different lambing rates, with 30 individuals of each breeds, were utilized for detection by the Sanger sequencing method to explore the polymorphism of the three specific mutations, as stated above.

### 2.3. Genotyping the Seven SNPs of the B4GALNT2 Gene Mentioned in WGS

A total of 99 experimental sheep from 10 breeds were used for WGS in our previous study, and seven SNPs in exons for the *B4GALNT2* gene were found. The primers for the seven SNPs (g.36971115C > T, g.36946470C > T, g.36946465G > A, g.36942215T > C, g.36933082C > T, g.36933070G > A, g.36930089T > G) detected were also designed based on the reference genome of ovis aries Oar_3.1 (Appendix A). Then, the polymorphisms of the seven SNPs for the *B4GALNT2* gene were also detected by the Sanger sequencing method in 88 STH sheep, which was recorded along with data of litter size. Finally, the relationship between the genotype frequency of the seven SNPs and litter size in STH sheep was analyzed by a chi-square test. The Linkage Disequilibrium (LD) pattern for the SNPs that were genotyped was plotted using Haploview (Version 4.2).

### 2.4. Amplification Full Length of Ovine B4GALNT2 Transcripts

Total RNA was extracted from each tissue using TRIzol Reagent (Invitrogen Inc., Carlsbad, CA, USA). cDNA for the tissue expression profile was synthesized using a PrimeScript^®^ RT Reagent Kit (Takara Bio Inc., Dalian, China) according to the manufacturer’s instructions. Total RNA from granulosa cells, which were cultured in vitro, was extracted for RACE (rapid amplification of cDNA ends)-Ready cDNA following the user manual of SMARTer^®^ RACE5′/3′ (Clontech Laboratories, Inc., Mountain View, CA, USA). The procedure of granulosa cells culture was as follows: Granulosa cells were aspirated from visible follicles (>3.0 mm in diameter), and separated from the follicular followed by washing with sterile Dulbecco’s Modified Eagle’s Medium (DMEM). The cells were evenly plated onto cell culture plates in the same medium supplemented with 15% fetal bovine serum and 1% penicillin streptomycin solution, and then incubated at 37 °C under 5% CO_2_ in humidified air.

There are several predicted transcripts for *B4GALNT2* gene in NCBI. In order to analyze the properties of B4GALNT2 protein subsequently, we should perform the cDNA clone to obtain the real and complete ovine *B4GALNT2* transcript sequence. The primers for the cDNA synthesis, shown in Appendix A, were designed according to the sequence of beta-1, 4-*N*-acetyl-galactosaminyl transferase 2 (*B4GALNT2*), transcript variant X7, mRNA, XM_012185950. P9–P14 were used for cDNA amplification and assembly (Appendix A). After the partial CDS sequence had been cloned, the 3’RACE GSP inner primer (3′GIP), 3′RACE GSP outer primer (3′GOP), and 5′RACE GSP primers (5′GP1-5′GP2) were designed to obtain the 3′UTR and 5′UTR of *B4GALNT2* (Appendix A). The procedure of touchdown Polymerase Chain Reaction (PCR) is as follows: Initial denaturation for 5 min at 94 °C, followed by 10 cycles of denaturation for 30 s at 94 °C, annealing for 30 s at 72 °C (with a decrease of 0.4 °C per cycle), and extension for 3 min at 72 °C, another 30 cycles of 30 s at 94 °C, 30 s at 68 °C, and 3 min at 72 °C, with a final extension for 8 min at 72 °C. The PCR products were purified using a DNA Purification Kit (TIANGEN Biotech Co., Ltd., Beijing, China), and were cloned into the pMDTM18-T Vector (Takara Bio Inc., Dalian, China), then transformed into competent cells DH5α (Takara Bio Inc., Dalian, China) for ultimate clone sequencing.

### 2.5. Bioinformatics Analysis

The DNA star was used to predict the *B4GALNT2* gene open reading frame (ORF) and amino acid sequence. According to the report of Wilkins et al., the fundamental characteristics of predicted B4GALNT2 proteins were forecasted by ProtParam (http://www.expasy.org/tools/protparam.html) [28]. The transmembrane domains were speculated by TMHMM (http://www.cbs.dtu.dk/services/TMHMM-2.0). The subcellular localization of the B4GALNT2 protein was assessed by TargetP (http://www.cbs.dtu.dk/services/TargetP/) and the Uniprot database (http://www.uniprot.org/). Potential disulphide bonds, *N*-glycosylation sites, and phosphorylation sites were predicted using the SCRATCH protein predictor (http://scratch.proteomics.ics.uci.edu/), NetNGlyc (http://www.cbs.dtu.dk/services/NetNGlyc/), and NetPhos (http://www.cbs.dtu.dk/services/NetPhos/), respectively. The hydrophilicity and average flexibility index of ovine B4GALNT2 amino acid sequences were analyzed by ProtScale (http://web.expasy.org/protscale/). SMART (Simple Molecular Architecture Reasearch Tool) (http://smart.embl-heidelberg.de/) and MEME (Multiple Em for Motif Elicitation) (http://meme-suite.org/tools/meme) were used to forecast the conserved domain of B4GALNT2. The secondary and tertiary structures of ovine B4GALNT2 were predicted by PSIPRED (http://bioinf.cs.ucl.ac.uk/psipred) and Phyre2 (http://www.sbg.bio.ic.ac.uk/phyre2/html/page. cgi?id=index), respectively. NCBI Blast (https://blast.ncbi.nlm.nih.gov/Blast.cgi) and MEGA software (version 6.0) were used to implement multiple alignments and molecular phylogenetic tree construction.

### 2.6. B4GALNT2 Expression Profile in STH Sheep and Tan Sheep

P15 in Appendix A was used to detect the expression profile of *B4GALNT2* in two sheep breeds by qRT-PCR (Quantitative Real Time Polymerase Chain Reaction). The housekeeping gene β-actin (P16) was adopted as the internal control in this study (Appendix A). A SYBR^®^ Premix Ex Taq™ kit (Takara Bio Inc., Dalian, China) for the qRT-PCR reaction was used in the LightCycler^®^ 480 Real-Time PCR system (Roche Applied Science, Mannheim, Germany) to quantify the expression levels of *B4GALNT2* in each tissue. Each 20 μL PCR reaction system contained 10 μL SYBR^®^ Premix Ex TaqTM II (Tli RNaseH Plus, 2×), 2 μL cDNA (ddH2O used as a blank control), 0.8 μL forward primer, and 0.8 μL reverse primer of the working solution concentration, and the rest of the volume was supplemented by ddH_2_O. The program of the qRT-PCR reaction is as follows: Denaturation at 95 °C for 3 min, followed by 40 cycles of 10 s at 95 °C, and 10 s at 60 °C, then the melting curves were collected. All the reactions were performed in triplicate. β-actin was used as the internal control, and the method of 2^−ΔΔCT^ was adopted to calculate the relative expression level of mRNA [29]. Analysis of Variance was performed for the significance test of expression analysis by SAS 8.0 software (SAS Institute Inc., Cary, NC, USA). Chi-square test was adopted to analyze the significant difference of litter size among the three genotype groups in each of the parities.

## 3. Results

### 3.1. Genotype Frequencies of Three Specific SNPs for the FecL Mutation

Three kinds of DNA fragments, which were used to detect the three SNPs closely associated with the *FecL* mutation sites (the mutation g.36938224T > A, localized in the intron 7 of *B4GALNT2*; the mutation g.37034573A > G, localized in the intergenic sequence between *B4GALNT2* and *EZR*; and the mutation *DLX3*:c. * 803A > G), were successfully amplified (Figure 1a–c). Three kinds of amplification fragment sizes were consistent with the anticipated lengths, which could be directly genotyped by the Sanger sequencing method. The results showed that, of the 11 measured sheep breeds, the consistency of high fecundity breeds (STH sheep, Hu sheep, Cele Black sheep) with low fecundity breeds (Tan sheep, White Suffolk sheep, Black Suffolk sheep, East Friesian sheep, Dorset sheep, Mutton Merino sheep, Dorper sheep, and Corriedale sheep) all had no variation in the three specific mutation sites (Figure 1d–f).

### 3.2. Genotype Frequencies for the Seven SNPs of the B4GALNT2 Gene in STH Sheep

Because none of the three known SNPs were identified in the eleven breeds, we looked up the data of WGS to search for other SNPs (sense mutation) in candidate gene (*B4GALNT2*). Fortunately, seven SNPs in *B4GALNT2* were found in the data of WGS. Combine with litter size data in STH sheep breed, the relationship between litter size and the unmentioned seven SNPs were studied in STH sheep. Seven SNPs in the exons of the ovine *B4GALNT2* gene discovered through WGS are listed in Table 1. Six of them had been found before and released in dbSNP with variant ID, and only the SNP in the position of g.36946465 was a new mutation. The four SNPs of g.36971115C > T, g.36946470C > T, g.36933082C > T and g.36930089T > G were detected in 88 STH ewes, which were recorded along with data of litter size. The CDS position of them were C205T, C482T, C865T and T1302G, respectively (Table 2).

### 3.3. SNPs of B4GALNT2 Associated with Litter Size in STH Sheep

For the four SNPs detected in STH sheep, the result of the chi-square test in Table 3 indicated that the genotype of g.36946470C > T and g.36933082C > T had a significant effect on litter size in the first parity (*p* < 0.05). In g.36946470C > T site, mean litter size of the first parity for CC and CT were 2.17 ± 0.11 and 2.32 ± 0.18, which were significantly higher than 1.25 ± 0.25 for the TT genotype (*p* < 0.05). However, the litter size between CC and CT showed no significant difference (*p* > 0.05). In the g.36933082C > T site, no TT genotype was found, and the mean litter size of the first parity for CC was 2.08 ± 0.09, which was significantly lower than 2.59 ± 0.24 for CT genotype (*p* < 0.05). In STH sheep, a haplotype block of *B4GALNT2* was identified: The block with the SNPs g.36971115C > T, g.36946470C > T, g.36933082C > T and g.36930089T > G (Appendix A). The site of g.36971115C > T and g.36930089T > G possessed a significant linkage disequilibrium with D′ > 0.8 and r^2^ > 0.3.

### 3.4. Cloning the Ovine B4GALNT2 cDNA Sequence

Based on reverse transcription cDNA from ovine granule cells , which were cultured in vitro, six fragments of *B4GALNT2* conserved sequences were successfully amplified using P4–P9 (Figure 2a–f). A sequence length of 2653 bp of the sheep *B4GALNT2* gene was assembled from the six partial conserved sequences. Then, 5′ and 3′UTR sequences were detected by the RACE method. Using two RACE primers (5′GP1–5′GP2), two transcription start sites (TSS) were discovered in 5′UTR (Figure 2g–h). Using the 3′RACE GSP inner primer (3′GIP) and 3′RACE GSP outer primer (3′GOP), a product of 924 bp based on *B4GALNT2* 3′UTR was cloned (Figure 2i). After sequencing and assembling the conserved sequences of 5′UTR and 3′UTR, a transcript of 3528 bp named *B4GALNT2a* and a transcript of 3514 bp named *B4GALNT2b* were submitted to NCBI. The accession number of *B4GALNT2a* (KY120333) and *B4GALNT2b* (KY120334) should be released on 16 November 2018.

### 3.5. Characterization of the Ovine B4GALNT2 cDNA Sequences

Genomic alignments indicated that the full-length cDNA sequence of *B4GALNT2a* and *B4GALNT2b* were all composed of 11 exons. Sequence analysis of *B4GALNT2a* and *B4GALNT2b* showed that they owned the same 3′UTR of 1924 bp and the same ORF of 1521 bp in size, which encoded for a 506 amino acid peptide. The difference between *B4GALNT2a* and *B4GALNT2b* was the length of 5′UTR; in other words, *B4GALNT2* possessed two transcription start sites (TSS), one of which was located at 83 bp, and the other of which that was located at 69 bp to the ATG (initiation codon) on the left (Figure 2j).

### 3.6. Feature and Structure Prediction of the Ovine B4GALNT2 Protein

Hydrophobicity analysis of the ovine B4GALNT2 protein indicated that the maximum hydrophobicity value was 3.167 in the position of 20 aa, and the minimum was −2.689 in the position of 483 aa (Figure 3a). The maximum and minimum average flexibility index values of ovine B4GALNT were 20.503 for the 121 aa position and 0.372 for the 416 aa position; however, the value of 0.379 for the 24 aa position is also worth pointing out and is discussed further below (Figure 3b). Then, SMART was used to speculate the conservative domains. There were a transmembrane region sequence at 9–31 aa, low complexity sequence at 117–128 aa, and Glycosyl transferase family 2 sequence at 264–421 aa, respectively (Figure 3c). The mutation of g.36933082C > T caused the residue change of P/S in the position of 288 aa, which belonged to the Glycosyl transferse family 2 domain. Therefore, this mutation should be considered more important than the other three mutations found in this study.

Secondary structural prediction showed that ovine B4GALNT2 is comprised of 25.49% alpha helix, 24.90% beta sheet, and 49.61% random coil (Appendix A). The tertiary structures of ovine B4GALNT2 were predicted by Phyre2. The results showed that the prediction model was based on a template of Nucleotide-diphospho-sugar transferases (d1xhba2), and only 243 residues (48% of the ovine B4GALNT2 sequence, 261–504 aa) have been modelled with 100.0% confidence by the single highest scoring template (Figure 3d).

### 3.7. Amino Acid Sequence Analysis of Ovine B4GALNT2

ProtParam was used to predict the physicochemical properties of amino acid sequences, and the molecular weight and isoelectric point of ovine B4GALNT2 were 57,443.4 Da and 8.86, respectively. The amino acid composition of the B4GALNT2 protein showed that the highest proportion of the B4GALNT2 protein was 12.1% for Leu, and the lowest was 1.0% for Trp.

The predictions of subcellular localization contribute to our understanding of the protein function and its physical and chemical environment in vivo [30]. Position 9–31 aa of the B4GALNT2 protein was predicted as being transmembrane helices (Figure 4a), and it was classified as a secretory pathway protein. Three disulphide bonds (cysteine pairs: 389–406, 453–504, and 79–81), two *N*-glycosylation sites (Asn232 and Asn305) and twenty-six phosphorylation sites (composed by ten Ser, eight Thr, and eight Tyr) were predicted in B4GALNT2 (Figure 4b,c).

### 3.8. Multiple Sequence Alignment and Phylogenetic Analysis

Six high similarity protein sequences (the ovine B4GALNT2 amino acid sequence was performed blastp in the NCBI based on the database of mammal’s Swiss-Prot for searching) were selected (Appendix A), and searched for the three most important motifs based on MEME (Figure 4d). In the amino acid sequence of ovine B4GALNT2, segments of 294–343 aa, 404–453 aa, and 206–255 aa were the most important regions for the motifs. The nucleotide sequence of B4GALNT2 in some models and domesticated animals were chosen in order to construct phylogenetic trees for the gene identified as homologous to ovine *B4GALNT2* (Appendix A). Then, a molecular phylogenetic tree was constructed by the Neighbor-Joining method, with 1000 Bootstrap replications in MEGA software (Version 6.0). At first glance, Figure 4e shows that the orthologous genes of ovine *B4GALNT2* in various species were clustered together first, paralogous genes were clustered later, and mammals were clustered versus *Xenopus tropicalis*.

### 3.9. B4GALNT2 Expression Profile in STH Sheep and Tan Sheep

To explore the differential expression patterns of *B4GALNT2* between high and low fecundity sheep breeds, qRT-PCR was performed on the cDNA of 13 tissues in STH sheep and Tan sheep. As shown in Figure 5, *B4GALNT2* was expressed in all of the detected tissues in the two sheep breeds. The *B4GALNT2* expression quantities in the ovary, kidney and lung were significantly higher than in the other tissues in STH sheep (*p* < 0.05); however, only the ovarian *B4GALNT2* expression quantity was significantly higher than that of other tissues in Tan sheep (*p* < 0.05). Comparing the expression in the same tissue between two sheep breeds, expression levels was significantly different in the kidney and oviduct (*p* < 0.05). As the fellow tissues of the HPG axis, the hypothalamus and pituitary were lower than other tissues in the expression of *B4GALNT2*.

## 4. Discussion

In France, the *FecL* mutation could be regarded as the best positional and expressional candidate for the high fecundity gene in Lacaune sheep breeds [17]. The female population of Lacaune sheep breeds is composed of 71% ++ ewes, 27% + L ewes, and 2% LL ewes. + L ewes produce 0.5 more lambs per lambing on average than ++ ewes [31]. In the present study, two mutations of g.36938224T > A and g.37034573A > G in intron of *B4GALNT2* and one mutation of *DLX3*:c. * 803A > G linked with the *FecL* site have not been found, which means that the difference in reproduction among these selected sheep breeds have nothing to do with the *FecL* mutation. However, two mutations of g.36946470C > T and g.36933082C > T in the exon of *B4GALNT2* had a significant effect on litter size in the first parity for STH Sheep (*p* < 0.05). Meanwhile, more data for the litter size of STH sheep should be collected to validate this conclusion.

Then, *B4GALNT2* cDNA was successively cloned from ovine granule cells, which were cultured in vitro. 5′RACE and 3′RACE were used to explore the full-length cDNA for the first time, and two transcription start sites (TSS) were obtained in 5′UTR, which meant alternative splicing existed in the *B4GALNT2* transcripts. Up to now, there has only been one transcript, which was included in NCBI. The transcription start sites (TSS) of 3514 bp fragments were the same as described in the research of Drouilhet et al. [17], which was in accordance with the transcript in NCBI. As early as 2003, human full-length clones of the *B4GALNT2* gene were obtained from the human colon cancer cell line Caco-2 [21,32]. Multiple transcripts diverged in their 5′ and 3′UTR, and some of them were very long (about 9000 bp), and the first exons existed as two alternative splices at least [33]. In many other glycosyltransferase genes, such as β1,4-galactosyltransferase, β4GT-I, and α2,3-sialyltransferase ST3Gal IV, multiple transcripts existed as well [34,35,36]. The transcriptional direction of ovine *B4GALNT2* was opposite to the established DNA sequence for mice, cattle, goats, pigs, and Rhesus monkeys, while the *B4GALNT2* transcriptional direction in humans and dogs was the exception. The two splicing isoforms of ovine *B4GALNT2* discovered in the present study were all contained in 11 exons, and had the same CDS of 1521 bp in size, which could encode a 506 amino acid peptide. To date, the verified human *B4GALNT2* transcripts in NCBI declared three alternative splicing fragments existed, and each length of its CDS was different produced with different lengths of residue. Therefore, there may be other alternative splicing fragments of ovine *B4GALNT2* gene, which remain to be found.

Based on the predicted data of subcellular localization, transmembrane helices, hydrophobicity, and average flexibility index of the B4GALNT2 protein, we suggest that the positions from 20 aa to 24 aa for the protein were possibly crucial for its normal function. The results of the transmembrane helices and conservative domains analysis showed that a transmembrane region existed in the position of 9–31 aa, and we can speculate that the ovine B4GALNT2 protein belonged to the pathway of secretory. According to the database of UniProt, the human B4GALNT2 protein was localized on the Golgi apparatus membrane [32,37]. Therefore, we can infer that the ovine B4GALNT2 protein may also be localized on the Golgi apparatus membrane. The B4GALNT2 protein in human transfers a beta-1,4-linked GalNAc to the galactose residue of an alpha-2,3-sialylated chain found on both *N*- and *O*-linked glycans [17,32]. In Lacaune sheep, DBA and KM694 staining were used to detect the different glycoprotein profiles, which were targeted by B4GALNT2 between L/L and +/+ ewes. Ten specific glycoproteins presented only in the follicular fluids of L/L ewes, including inhibin α and βA subunits, which lead to the production of Activin A and Inhibin A and finally influence the ovarian function [17].

Based on homology modeling, the tertiary structure prediction of the ovine B4GALNT2 protein only covered the position of 261–504 aa, and the result was consistent with the speculation of conservative domains, in which a Glycosyl transferse family 2 sequence was located in the position of 264–421 aa. The positions of 416 aa (the minimum average flexibility index), 483 aa (the minimum value of hydrophobicity), and 389 aa and 406 aa (a pair of disulphide bonds) were marked on the tertiary structure. 453 aa and 504 aa should have also made up a pair of disulphide bonds based on the SCRATCH protein predictor (http://scratch.proteomics.ics.uci.edu/); however, their positions were far away from each other in the predicted tertiary structure. As it turned out, the evidence suggests that tertiary structure prediction in the present study offered limited utility. After multiple sequence alignment of the amino acid sequence, six high similarity sequences analyzed with MEME revealed that the three most important motifs were located in the sequence of the Glycosyl transferse family 2, which verified the speculation of the conservative domain.

Although the two linked mutations (mutation g.36938224T > A and mutation g.37034573A > G) were not located in the exons of *B4GALNT2*, Drouilhet et al. took into account the discovery that the *B4GALNT2* gene expressed in L/L granulosa and theca cells was exhibited at a 1000 fold higher rate than in +/+, and this finally supported the *B4GALNT2* gene as the candidate gene for *FecL* [17]. The new viewpoint of the *B4GALNT2* gene acting as the *FecL* gene was announced widely by researchers for the reason that this gene affects litter size in a different way, which have nothing to do with the TGFβ/BMP signal pathway [1,17,38,39]. The mutation of g. 36946470C > T caused the residue change of P/L in the positing of 160aa, the position of which has no important domain for the protein. However, the mutation of g.36933082C > T caused the residue change of P/S in the position of 288 aa, and the position belonged to the Glycosyl transferse family 2 domain (264–421 aa). Therefore, this mutation of g.36933082C > T should be considered more important than the mutation of g. 36946470C > T in this study.

In the present study, the expression quantities of *B4GALNT2* in the ovary were all significantly higher than those of other tissues in STH sheep and Tan sheep, which was consistent with the results of Drouilhet et al. [17]. Except for the kidney and oviduct, no significant differences in expression level was found between the two breeds in the other tissues. The expression results suggest that the *B4GALNT2* gene should play an important role in the ovine ovary, while having little correlation with differences between STH sheep and Tan sheep in reproduction. Li et al. reported that *B4GALNT2* is expressed in the colon, kidney, intestine, oviduct, and ovary. *B4GALNT2* is up-regulated by progesterone and down-regulated by estrogen in mice, and progesterone-induced up-regulation of *B4GALNT2* might contribute to the implantation process [40,41]. The mice B4GALNT2 protein was detected in the whole cumulus-oocyte, and Lin had proven that the B4GALNT2 protein was essential for cumulus expansion, which was a critical step during oogenesis [42]. The B4GALNT2 protein was localized in the granulosa cells and the antral follicular fluid of Lacaune sheep (L/L) ovaries as well, and performed the transfers of beta-1,4-linked GalNAc to the galactose residue of an alpha-2,3-sialylated chain [32]. Mice with oocytes generating glycoproteins lacking core 1-derived *O*-glycans and complex *N*-glycans had been regarded as a new model of follicular premature ovarian failure [43]. These findings proved the vital importance of glycosylation in the control of ovarian function, and pointed to a new direction of discovery in the gene pathway of ovine reproduction.

## 5. Conclusions

In summary, the selected 11 sheep breeds had no variation in the three specific mutation sites, which are closely linked to *FecL* mutation. However, two mutations of g.36946470C > T and g.36933082C > T in the exon of *B4GALNT2* had a significant effect on litter size in the first parity for STH Sheep (*p* < 0.05). Two transcription start sites (TSS) were discovered from 5′UTR of *B4GALNT2* in ovine granule cells in vitro. *B4GALNT2* is mainly expressed in the ovine ovary, suggesting that *B4GALNT2* plays an important role in sheep reproduction. The pathway of *B4GALNT2*, which is different from the BMP pathway, is a new direction of research to further understand ovine ovarian function. Our findings and analysis of ovine *B4GALNT2* will help us to further understand its expression and function, and also may contribute to exploring its role in the ewe reproduction system.

## Figures and Tables

**Figure 1 animals-08-00160-f001:**
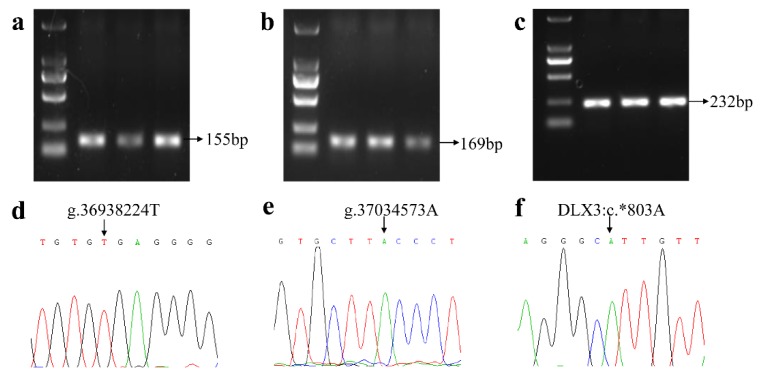
Target amplification bands for three specific mutations. (**a**) Target band amplified by P1 for the mutation g.36938224T > A. (**b**) Target band amplified by P2 for the mutation g.37034573A > G. (**c**) Target band amplified by P3 for the mutation DLX3:c. * 803A > G. (**d**) The sequencing result at the site of g.36938224T > A. (**e**) The sequencing result at the site of g.37034573A > G. (**f**) The sequencing result at the site of DLX3:c. * 803A > G.

**Figure 2 animals-08-00160-f002:**
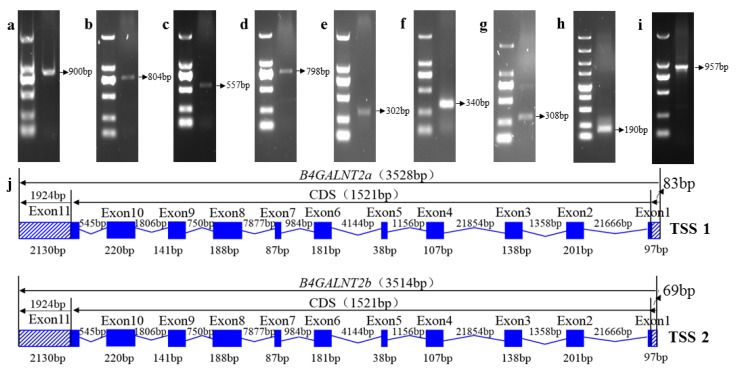
Full-length sequence of *B4GALNT2*. (**a**) Partial cording sequence amplified by P4. (**b**) Partial cording sequence amplified by P5. (**c**) Partial cording sequence amplified by P6. (**d**) Partial cording sequence amplified by P7. (**e**) Partial cording sequence amplified by P8. (**f**) Partial cording sequence amplified by P9. (**g**) 5′RACE product for the transcript of *B4GALNT2a*. (**h**) 5′RACE product for the transcript of *B4GALNT2b*. (**i**) 3′RACE product for the *B4GALNT2* gene. (**j**) Gene structures of the *B4GALNT2a* and *B4GALNT2b* transcripts. RACE, rapid amplification of cDNA ends.

**Figure 3 animals-08-00160-f003:**
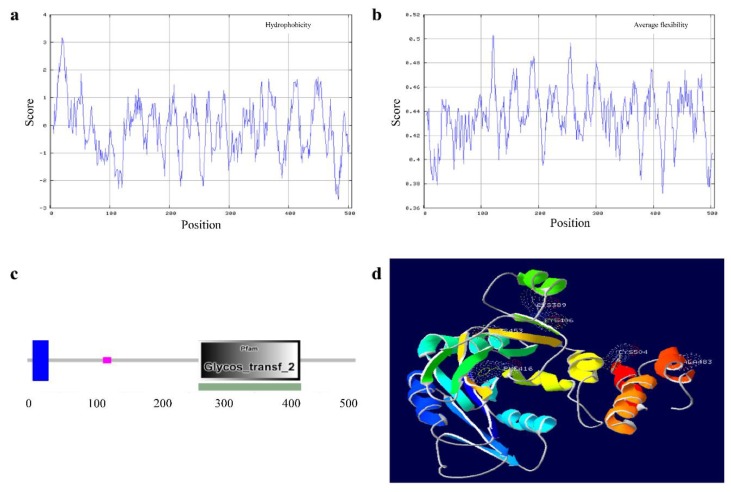
Feature and structure prediction of the ovine B4GALNT2 protein. (**a**) Hydrophobicity analysis of the B4GALNT2 protein. (**b**) Average flexibility index of the B4GALNT2 protein. (**c**) Predicted conservative domains of the B4GALNT2 protein. (**d**) Prediction of partial sequence for B4GALNT2 in tertiary structures.

**Figure 4 animals-08-00160-f004:**
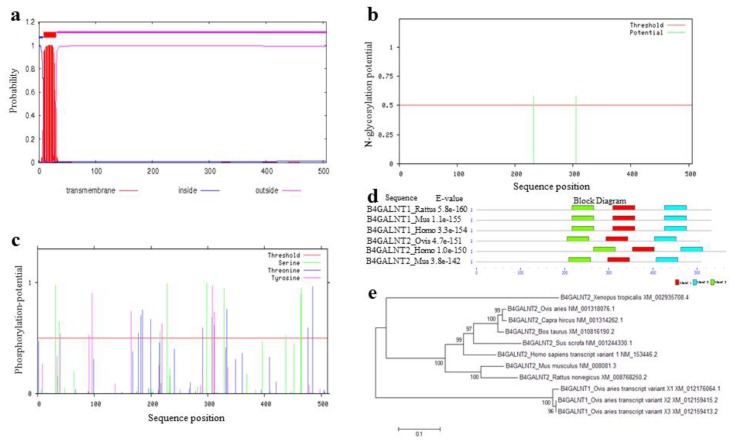
Amino acid sequence analysis of ovine B4GALNT2 and its multiple sequence alignments. (**a**) Predicted transmembrane helices of the B4GALNT2 protein. (**b**) Predicted *N*-glycosylation sites of the B4GALNT2 protein. (**c**) Predicted phosphorylation sites of the B4GALNT2 protein. (**d**) Prediction of the three most important motifs in the B4GALNT2 protein. (**e**) Phylogenetic tree based on the homology nucleotide sequence for B4GALNT2.

**Figure 5 animals-08-00160-f005:**
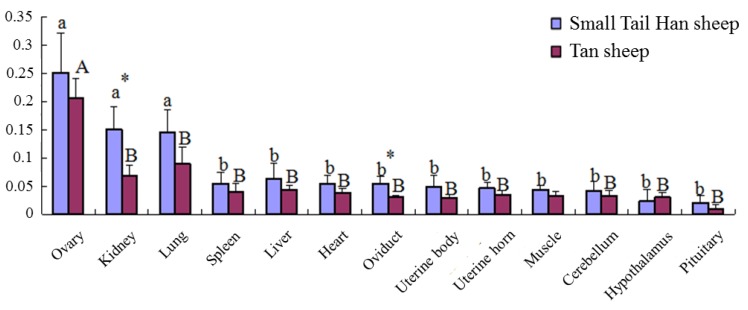
*B4GALNT2* expression profile in STH sheep and Tan sheep. Note: Lowercase letters indicate significant differences among tissues in Small Tail Han sheep, and capital letters indicate differences among tissues in Tan sheep. Means with the same letter were not significantly different (*p* < 0.05). The symbol of “*” indicates significant differences (*p* < 0.05) between STH sheep and Tan sheep in same tissues.

**Table 1 animals-08-00160-t001:** SNPs for missense in the ovine *B4GALNT2* gene.

Chromosome	Position	Codons	Residues Change	Residues Position	Source	Variant ID
chr11	g.36971115	CAC/TAC	H/Y	68	missense	rs595977899
	g.36946470	CCA/CTA	P/L	160	missense	rs426776354
	g.36946465	GCT/ACT	A/T	162	missense	NA
	g.36942215	ATT/ACT	I/T	197	missense	rs423653795
	g.36933082	CCA/TCA	P/S	288	missense	rs405227267
	g.36933070	GTG/ATG	V/M	292	missense	rs426513123
	g.36930089	CAT/CAG	H/Q	433	missense	rs402747554

Note: NA indicates Not Applicable to its column.

**Table 2 animals-08-00160-t002:** Allele and genotype frequencies of *B4GALNT2* in Small Tail Han (STH) ewes.

Polymorphic Site (CDS Position)	Genotype	Genotype Frequency (N)	Allele	Allele Frequency	χ^2^ (*p*)
g.36971115C > T (C205T)	CC	0.85 (74)	C	0.92	0.40 (0.53)
	CT	0.14 (12)	T	0.08	
	TT	0.11 (1)			
g.36946470C > T (C482T)	CC	0.68 (59)	C	0.82	0.78(0.38)
	CT	0.27 (23)	T	0.18	
	TT	0.05 (4)			
g.36933082C > T (C865T)	CC	0.81 (71)	C	0.9	1.01 (0.32)
	CT	0.19 (17)	T	0.1	
	TT	0 (0)			
g.36930089T > G (T1302G)	TT	0.69 (61)	T	0.82	0.87 (0.35)
	TG	0.26 (23)	G	0.18	
	GG	0.05 (4)			

Note: *p* > 0.05 indicates the locus was under Hardy-Weinberg equilibrium.

**Table 3 animals-08-00160-t003:** Litter size and standard error of STH ewes in different parities for each genotypes.

Polymorphic Site	Genotype	Litter Size (Means ± S.E.)			
		First Parity (N)	Second Parity (N)	Third Parity (N)	Total (N)
g.36971115C > T	CC	2.18 ± 0.1 (68)	2.37 ± 0.13 (52)	2.7 ± 0.22 (23)	2.43 ± 0.08 (144)
	CT	2.36 ± 0.2 (11)	2.43 ± 0.37 (7)	3 ± 0 (3)	2.60 ± 0.23 (21)
	TT	1 ± 0 (1)	2 ± 0 (1)	2.00 ± 0 (1)	1.67 ± 0.52 (3)
g.36946470C > T	CC	2.17 ± 0.11 (54) ^b^	2.32 ± 0.16 (38)	2.87 ± 0.27 (15)	2.45 ± 0.10 (107) ^b^
	CT	2.32 ± 0.18 (22) ^b^	2.37 ± 0.19 (19)	2.6 ± 0.26 (10)	2.43 ± 0.13 (51) ^b^
	TT	1.25 ± 0.25 (4) ^a^	2.33 ± 0.333(3)	2.00 ± 1 (2)	1.58 ± 0.38 (7) ^a^
g.36933082C > T	CC	2.08 ± 0.09 (64) ^a^	2.32 ± 0.14 (47)	2.57 ± 0.2 (21)	2.32 ± 0.09 (132) ^b^
	CT	2.59 ± 0.24 (17) ^b^	2.5 ± 0.25 (14)	3.29 ± 0.42 (7)	2.79 ± 0.15 (38) ^a^
	TT	NA	NA	NA	NA
g.36930089T > G	TT	2.11 ± 0.12 (55)	2.32 ± 0.15 (41)	2.8 ± 0.25 (20)	2.41 ± 0.09 (116)
	TG	2.43 ± 0.15 (23)	2.5 ± 0.2 (18)	2.5 ± 0.22 (6)	2.48 ± 0.15 (47)
	GG	1.67 ± 0.33 (3)	2 ± 0 (2)	3.00 ± 1 (2)	2.22 ± 0.35 (7)

Note: Values with the same superscript for the same column have no significant difference (*p* > 0.05). Values with a different superscript for the same column have significant differences (*p* < 0.05).

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
