# Peer review of "Molecular Cloning of the B4GALNT2 Gene and Its Single Nucleotide Polymorphisms Association with Litter Size in Small Tail Han Sheep"

_animals, 2018, doi:10.3390/ani8100160_

Round 1

Reviewer 1 Report

English has to be revised as it is difficult to follow and many sentences lack meaning.

The study on the role of different polymorphisms in two genes for the trait litter size or fecundity rate in ewes of different breeds concludes that the Lacaune mutation in B4GALNT2 is not associated to the higher litter size in the STH Chinese breed. However, other mutations are seen to be linked to this trait and authors make a hard work to characterise the protein in different tissues. Finally there is no big conclusion as litter size in STH is partially explained by the polymorphism in this gene but signification is low and only present in first parity.

The manuscript is difficult to follow: what happens with the eleven breeds used as only STH is mentioned thereafter. The different steps are hard to follow from material and methods to results and this should be restructured to help understanding.

Line 64. The sentence is contradictory as the SNP found in Lacaune and identified as the FecB mutation in B4GALNT2 has not been identified in other breeds yet but other mutations responsible of fertility rate in ewes and positioned in other genes as mentioned by the authors. Rewrite the sentence.

Why cloning the transcript when authors have all the sequence information (previous NGS) to produce it by RT PCR with no need of RACE technology?

Granulosa cells culture? Where is this mentioned in material and methods

Why finally genotyping only three SNPs instead of the 7 identified in B4GALNT2 and the one in DLX3?

Table 3: what is the number in parenthesis? Number of ewes with the genotype?

The homology tree in figure 4 e is surprising; the three variants found in the manuscript are completely distant from the other O.aries sequences which are nearer from porcine bovine  and mouse then the three new sequences! This would mean some important errors in the sequence which make spurious nucleotide variation to happen. Please have a look to that.

I recommend to revise the manuscript, make it clearer more concise and rewrite the text with the help of an english specialist. 

Author Response

Reviewer 1: 

1. English has to be revised as it is difficult to follow and many sentences lack meaning.

Response:

Thank you for showing our errors in the written English language. We have tried our best to revise the manuscript sentence by sentence. A professional editing company (MDPI English Editing Service) also helped us with language revision in the revised version.

2. The manuscript is difficult to follow: what happens with the eleven breeds used as only STH is mentioned thereafter. The different steps are hard to follow from material and methods to results and this should be restructured to help understanding.

Response:

The eleven breeds were used to detected the three known SNPs. However, none of the three known SNPs were identified in the eleven breeds. Then, we looked up the data of WGS to search for other SNPs (sense mutation) in candidate gene (B4GALNT2). Fortunately, 7 SNPs in B4GALNT2 were found in the data of WGS. Because we only had the litter size data in STH sheep breed, hence, the relationship between litter size and the unmentioned 7 SNPs were studied in STH sheep.

We had added some statements in results (3.2. Genotype Frequencies for the Seven SNPs of the B4GALNT2 Gene in STH Sheep) to make our manuscript easier to understand.

3. Line 64. The sentence is contradictory as the SNP found in Lacaune and identified as the FecB mutation in B4GALNT2 has not been identified in other breeds yet but other mutations responsible of fertility rate in ewes and positioned in other genes as mentioned by the authors. Rewrite the sentence.

Response:

We rewrite the sentence of.Line 64. “FecB gene was found in Booroola Merino sheep and regarded as the first major gene for prolificacy, which was also identified in other various sheep breeds [7, 22-25]. However, whether the effect of FecL mutation which found in Lacaune sheep breed was also exist in other sheep breeds remains unknown.” is corrected as “The FecB gene was found in Booroola Merino sheep and regarded as the first major gene for prolificacy, and was also identified in various other sheep breeds [7,22–25]. However, whether the effects of the FecL mutation which was found in the Lacaune sheep breed also exists in other sheep breeds remains unknown.”

4. Why cloning the transcript when authors have all the sequence information (previous NGS) to produce it by RT PCR with no need of RACE technology?

Response:

    The whole-genome sequencing (WGS) only provided information of DNA sequence; we did not know about the sequence information in RNA transcriptional level. For example, how many transcripts for B4GALNT2, which exon is easy to lose in mRNA, where is the transcription start site in the sequence? These questions can be answered with RACE technology.

5. Granulosa cells culture? Where is this mentioned in material and methods

Response:

The brief introduction of granulosa cells culture was added into the material and methods as follows:

Granulosa cells were aspirated from visible follicles (>3.0 mm in diameter), and separated from the follicular followed by washing with sterile Dulbecco’s Modified Eagle’s Medium (DMEM). The cells were evenly plated onto cell culture plates in the same medium supplemented with 15% fetal bovine serum and 1% penicillin streptomycin solution and then incubated at 37 °C under 5% CO2 in humidified air.

6. Why finally genotyping only three SNPs instead of the 7 identified in B4GALNT2 and the one in DLX3?

Response:

The three SNPs (g.36938224T>A, g.37034573A>G of B4GALNT2 and DLX3:c.*803A>G) are known SNPs that have been reported in previous research. In 2009, researchers conclude that the DLX3:c.*803A>G SNP provided accurate classification of 99.5% sheep as carriers or non-carriers of the FecL mutation. Four years later, consideration of two mutations (the SNP g.36938224T>A, localized in the intron 7 of B4GALNT2, and the SNP g.37034573A>G localized in the intergenic sequence between B4GALNT2 and EZR) that were closely associated with the FecL mutation; researches thought B4GALNT2 appeared as the best positional and expressional candidate for FecL gene.

Therefore, the three known SNPs marked for FecL were genotyping first in the selected 11 sheep breeds. While, any one of the three known SNPs was not identified. Then, we looked up the data of WGS to search for other SNPs (sense mutation) in candidate gene (B4GALNT2). Fortunately, 7 SNPs in B4GALNT2 were found in the data of WGS. Hence, the relationship between litter size and the unmentioned 7 SNPs were studied in this manuscript.

7. Table 3: what is the number in parenthesis? Number of ewes with the genotype?

Response:

    Thank you for your reminder. The number in parenthesis is the number of ewes with the genotype. We labeled (N) in the header of the table.

8. The homology tree in figure 4 e is surprising; the three variants found in the manuscript are completely distant from the other O.aries sequences which are nearer from porcine bovine and mouse then the three new sequences! This would mean some important errors in the sequence which make spurious nucleotide variation to happen. Please have a look to that.

Response:

    You may be mistaken. The three variants on the bottom were all about the gene of B4GALNT1 not the gene of B4GALNT2. So there is no error in this figure.

9. I recommend to revise the manuscript, make it clearer more concise and rewrite the text with the help of an english specialist.

Response:

    Thank you for your advice. We have tried our best to revise the manuscript according to comments of reviewers. Moreover, an English specialist in MDPI English Editing Service polishes our manuscript. 

Reviewer 2 Report

     FecL is a fecundity gene recently found in French Lacaune sheep. Three SNPs, DLX3:c*803A>G, g.36938224T>A, g.37034573A>G were known to be closely associated with the FecL mutation. In 2009, Drouilhet et al. revealed that beta-1,4- N- acetyl-galactoaminyl transferase 2 (B4GALNT2) is the potential FecL gene. They also showed that glycoprotein profiles in the follicular fluids were different between in L/L and +/+ ewes.

     In ths study, Guo et al, revealed the following things,

1) Three known SNPs associated with FecL gene were know found 11 sheep breed other than Flench Lacaune sheep.

2) Two SNPs located in exson of B4GALNT2 had a significant effect on litter size in the frrst parity for Small Tail Han (STH) sheep.

3) Full length cDNA sequence of ovine B4GALNT2 were cloned for the first time, and two transcription start sites were discovered.

4) B4GALNT2 was mainly expressed in ovine ovary, indicating that B4GALNT2 plays an important role in sheep production.

Full length cDNA of B4GALNT2 have been already cloned in human, and effect of B4GALNT2 on fecundity traits have been studied in French Lacaue sheep. So, I thought that the novelty and the scientific significance of these results were low, but this study showed that B4GLNT2 mutation affected the fecundity traits in a sheep breed other than French Lacaune sheep for the first time, and has some value to be published.

I recommend to the authors to improve the following points.

1. At the time of reading the abstract, I did not understand why the cDNA cloning of B4GLNT2 had to be done. So I recommend the authors to describe clearly the necessity of the cDNA cloning.

2. I would like the authors to deeply discuss about the potential effect of two SNPs, g.36946470C>T and g.36933082C>T on the enzymatic activity of B4GLNT2 or their genetic conservation.

3. In Figure 2j, I think that it is better to express more emphasized two different transcriptional start sites.

4. In many figures, the resolution was low and characters were small, so it was hard to understand. I hope the figures are improved.

Author Response

Reviewer 2: 

1. At the time of reading the abstract, I did not understand why the cDNA cloning of B4GLNT2 had to be done. So I recommend the authors to describe clearly the necessity of the cDNA cloning.

Response:

To emphasize the necessity of the cDNA cloning, we add some statement in the part of materials and methods (2.4 Amplification Full Length of Ovine B4GALNT2 Transcripts). The statement is as follows:

“There are several predicted transcripts for B4GALNT2 gene in NCBI. In order to analyze the properties of B4GALNT2 protein subsequently, we should perform the cDNA clone to obtain the real and complete ovine B4GALNT2 transcript sequence.”

2. I would like the authors to deeply discuss about the potential effect of two SNPs, g.36946470C>T and g.36933082C>T on the enzymatic activity of B4GLNT2 or their genetic conservation.

Response:

The statements of “The mutation of g. 36946470C>T caused the residue change of P/L in the positing of 160aa, which the position have no important domain for the protein. However, the mutation of g.36933082C>T caused the residue change of P/S in the position of 288aa, and the position belonged to the Glycosyl transferse family 2 domain (264-421a). Therefore, this mutation of g.36933082C>T should be considered more important than the mutation of g. 36946470C>T in this study.” was added at the fifth paragraph in the discussion.

3. In Figure 2j, I think that it is better to express more emphasized two different transcriptional start sites.

Response:

Thank you for your kindly advice. We replaced the Figure 2j, and we labeled the two TSS of B4GALNT2 in the figure.

4. In many figures, the resolution was low and characters were small, so it was hard to understand. I hope the figures are improved.

Response:

Thank you for your kindly advice. We made the characters in figures bigger, and the high-resolution figure in form of PDF version was also submitted.

Reviewer 3 Report

No comments and suggestions.